# Metabolomic Profiling in Neuromyelitis Optica Spectrum Disorder Biomarker Discovery

**DOI:** 10.3390/metabo10090374

**Published:** 2020-09-18

**Authors:** Maxton E. Thoman, Susan C. McKarns

**Affiliations:** 1Department of Surgery, University of Missouri School of Medicine, Columbia, MO 65212, USA; metkcc@health.missouri.edu; 2Laboratory of TGF-β Biology, Epigenetics, and Cytokine Regulation, Department of Surgery, University of Missouri School of Medicine, Columbia, MO 65212, USA; 3Department of Microbiology and Immunology, University of Missouri School of Medicine, Columbia, MO 65212, USA

**Keywords:** astrocyte, AQPA4, NMO, metabolomics, SCFA, lactate

## Abstract

There is no specific test for diagnosing neuromyelitis optica spectrum disorder (NMOSD), a disabling autoimmune disease of the central nervous system. Instead, diagnosis relies on ruling out other related disorders with overlapping clinical symptoms. An urgency for NMOSD biomarker discovery is underscored by adverse responses to treatment following misdiagnosis and poor prognosis following the delayed onset of treatment. Pathogenic autoantibiotics that target the water channel aquaporin-4 (AQP4) and myelin oligodendrocyte glycoprotein (MOG) contribute to NMOSD pathology. The importance of early diagnosis between AQP4-Ab^+^ NMOSD, MOG-Ab^+^ NMOSD, AQP4-Ab^−^ MOG-Ab^−^ NMOSD, and related disorders cannot be overemphasized. Here, we provide a comprehensive data collection and analysis of the currently known metabolomic perturbations and related proteomic outcomes of NMOSD. We highlight short chain fatty acids, lipoproteins, amino acids, and lactate as candidate diagnostic biomarkers. Although the application of metabolomic profiling to individual NMOSD patient care shows promise, more research is needed.

## 1. Introduction

The term ‘neuromyelitis optica spectrum disorder’ (NMOSD) was established as a way to unify and classify neuromyelitis optica (NMO), formerly known as Devic’s disease, and a variety of related neurodegenerative syndromes and autoimmune disorders in order to improve individual patient care [1]. There is no cure for NMOSD. Our understanding of the complexity of NMOSD pathology has grown over the past decade. Yet, there remains no definitive discriminatory biomarker for early disease diagnosis. The probability that the continued use of single-candidate approach studies, that may not capture the dynamic complexity of the underlying cause(s) of NMOSD or mechanism(s) of disease progression is limited. Metabolomic profiling provides a means to quantitatively compare thousands of endogenous metabolites from multiple tissues at multiple timepoints over the course of disease from multiple subjects. The objective of this review is to provide a comprehensive data collection and analysis of the currently known metabolomic perturbations and related proteomics outcomes of NMOSD and relate these immune cell function. Our hope is that this extensive data resource will facilitate additional metabolomic profiling to expedite biomarker discovery. An individual’s metabolomic profile is reportedly as unique as one’s fingerprint given that no two individuals have exactly the same enzyme activity at any given time—and thus holds great promise to facilite precision medicine approaches for NMOSD patient care.

### 1.1. A Role for AQPA-IgG_1_ and MOG-IgG_1_ Autoantibodies in NMOSD Pathology

NMOSD characteristically involves the inflammation and demyelination of the optic nerves (optic neuritis) and spinal cord (transverse myelitis), and less commonly affects the brainstem and hypothalamus to cause pain, blindness, and paralysis [2,3,4]. Characteristic lesions, including recurrent attacks of optic neuritis and longitudinally extensive transverse myelitis (LETM) by magnetic resonance imaging (MRI), are required to fulfill NMOSD diagnosis. The onset of optic neuritis in NMOSD is believed to trigger subsequent inflammation and swelling, which increases the likelihood of pain and the loss of vision. Alternatively, spinal cord damage triggers subsequent inflammatory responses that potentiate fatigue, weakness, paralysis, and sensory, bowel, and bladder dysfunction. Nearly 90% of patients undergo disease remission and relapse (known as relapsing NMOSD), while the remaining patients develop progressive monophasic NMOSD [5]. 

Immunoglobulin G_1_ (IgG_1_) autoantibodies (Ab) that target the astrocyte aquaporin-4 (AQP4) water channel have been detected in approximately 70% of NMOSD patients worldwide [6,7,8,9]. However, this estimate may be underestimated, because not all testing used the more sensitive diagnostic AQP4 cell-based assay (CBA) [10,11]. AQP4-Ab binds to and internalizes AQP4 expressed on astrocyte end feet surrounding blood vessels and synapses [4,6,12,13,14,15]. Immunoglobulin G_1_ (IgG_1_) autoantibodies that target myelin oligodendrocyte glycoprotein (MOG-Ab) has been detected in approximately 40% of AQP4-Ab^−^ NMOSD patients [16,17,18,19,20,21]. MOG-Ab binds to MOG and destroys myelin covering neurons. AQP4-Ab and MOG-Ab further contribute to disease pathology by fixing complement, which results in astrocyte and neuron death. Neither AQP4-Ab or MOG-Ab have been detected in the remaining NMOSD patients. The presence of both AQP4-Ab and MOG-Ab have not been detected in any NMOSD patients. The influx of neutrophils and eosinophils into the CNS plays a key role in propagating inflammation and cell damage [12,13,14,15,22,23,24]. More recently, AQP4-reactive CD4^+^ T cells and gut microbiome dysbiosis have also been linked to disease pathology [25]. 

### 1.2. The Urgency for Novel NMOSD Biomarker Discovery 

Relapsing NMOSD rarely converts to a secondary progressive phenotype to support the opinion that the underlying mechanism(s) that cause these two disease subtypes are different [26]. Current treatment options for NMOSD include a variety of non-specific immune suppressive modalities [27,28,29,30,31,32,33,34,35,36]. The more recent inclusion of B cell-depleting monoclonal antibodies has improved patient response to therapy, given the importance of humoral immunity in driving disease pathology [36,37,38,39]. Unfortunately, neither AQP4 or MOG autoantibodies predict disease relapse, response to therapy, or prognosis. Promising new therapies, such as eculizumab (C5 complement inhibitor), tocilizumab (interleukin (IL)-6 receptor blocker), and Cl-esterase inhibitor (complement Clr and Cls inhibitor) [28,29,40,41,42], bring new options for treatment. However, the continued absence of biomarkers for early diagnosis between AQP4-IgG^+^ NMOSD, AQP4-IgG^−^ MOG-IgG^+^ NMOSD, AQP4-IgG^−^ MOG-IgG^−^ NMOSD, and related disorders, therapy will remain hindered. An urgency for novel biomarker discovery is further highlighted by the challenge to discriminate between MOG-IgG_1_^+^ NMOSD and related MOG antibody disease (MOGAD) patients that are also MOG-IgG_1_^+^ [16,17,18,19,20,21] and between AQP4-IgG^−^ MOG-IgG^−^ NMOSD and related AQP4-IgG^−^ MOG-IgG^−^ disorders with overlapping clinical symptoms. The presence of AQP4-IgG unequivocally differentiates NMOSD from multiple sclerosis [6,7,8,9]. Increased cerebrospinal fluid (CSF) eosinophils, neutrophils, glial fibrillary acidic protein (GFAP), or neurofilament light chain and the absence of CSF-restricted oligoclonal IgG bands also argue in favor of NMOSD over multiple sclerosis [43,44,45]. The misdiagnosis of NMOSD may result in adverse effects to therapies by NMOSD patients [46,47,48,49], and highlight the importance of early diagnosis and monitoring of individual NMOSD patient response to therapy. 

### 1.3. Metabolomic Profiling Applied to Autoimmune Disease 

Metabolic processes, including glycolysis, the tricarboxylic acid (TCA) cycle, the pentose phosphate pathway, fatty acid oxidation, fatty acid synthesis, and amino acid metabolism, are essential for precise regulation of immune cell fate. These pathways are tightly regulated by a host of factors, including energy, nutrients, cytokines, chemokines, and microbes and their metabolites. Metabolic perturbations of specific metabolites have been implicated in autoimmune disease. For example, ultra-high-performance liquid chromatography coupled with high-resolution mass spectrometry (UPLC-HRMS) has differentiated rheumatoid arthritis from primary Sjogren’s syndrome [50]. These are two autoimmune disorders that present with overlapping symptoms of joint pain, swelling, and stiffness. In this study, rheumatoid arthritis had increased levels of serum 4-methoxyphenylacetic acid, glutamic acid, l-leucine, l-phenylalanine, l-tryptophan, l-proline, glyceraldehyde, formate, and cholesterol, but decreased levels of capric acid, argininosuccinic acid, and bilirubin when compared to Sjogren’s syndrome patients. In pediatric type 1 diabetes mellitus, the presence of islet autoantibodies associates with increased phospholipids and triglycerides, but decreased methionine and hydroxyproline levels [51]. The discovery of metabolic dysbiosis in systemic lupus erythematosus (SLE) has identified novel targets for drug development [52,53]. For instance, metformin, an anti-diabetic drug that regulates systemic and cellular metabolism, promotes CD4^+^ T regulatory cell expansion [54]. Moreover, metabolomic profiling of celiac disease has led to the inclusion of personalized lifestyle and dietary habits as prophylactic intervention for individuals suffering from gluten intolerance [55,56].

### 1.4. Immunometabolism in Autoimmune Disease

Immunometabolism, an emerging frontier that investigates the interplay between immunological and metabolic processes, is gaining momentum due to the growing appreciation that dysregulated metabolic processes underlie many aberrant immune-mediated responses. As such, cellular metabolism is a target for drug development. The environment contributes to autoimmune disease pathology, and the same principle holds true for metabolism. Tissue metabolism shifts, in part, following the recruitment of inflammatory cells, including neutrophils and monocytes, and local expansion of effector lymphocytes [57]. As lymphocytes transition from a resting to proliferative state, they undergo a metabolic switch in order to acquire substrates from the environment in order to meet their increased energy demands required for effector function [58]. Immune cells with different functions use different metabolic pathways. Although these metabolic pathways rely on the same pool of fuel sources and key synthetic precursors, they generate different outputs in different cell types. The importance of understanding that these needs change markedly between states of inflammation and disease relapse and not be overemphasized.

### 1.5. A Role for Metabolomics in Multiple Sclerosis 

Metabolomic profiling has been used in order to discriminate between multiple sclerosis and other related diseases, discriminate between RRMS, secondary progressive multiple sclerosis (SPMS), and primary progressive multiple sclerosis (PRMS), and to characterize patient disability [59,60,61,62,63,64,65,66,67,68,69,70,71,72,73,74,75,76,77,78,79,80]. These studies have incorporated human subjects and experimental animal models. Serum samples from multiple sclerosis patients have differentiated from healthy controls by increased levels of serum sphingomyelin, phosphatidyl-ethanolamine, phosphatidyl-choline, phosphatidyl-inositol, glutamate, and selective amino acids, as determined by mass spectrometry (MS) and nuclear magnetic resonance (NMR) spectroscopy [59,63,67,76]. The serum levels of sphingomyelin, phosphatidyl-ethanolamine, phosphatidyl-choline, phosphatidyl-inositol, glutamate, and other amino acids have further been identified as possible biomarkers to discriminate between multiple sclerosis and healthy controls, as determined by UHPLC-MS and UHPLC-TOF-MS. Further, SPMS has been differentiated from RRMS by decreased levels of phosphatidylcholine, N-acetyl species, lactate, and glucose, but increased levels of other fatty acids and β-hydroxybutyrate [81]. 

Multiple sclerosis has discriminated from NMOSD by increased levels of serum lysine and histidine and decreased large HDL particles, unsaturated lipid, and alanine in a lipid profiling study [82]. When comparing CSF profiles between multiple sclerosis and healthy controls, multiple sclerosis had increased levels of choline, myo-inositol, and threonate, but decreased levels of 3-hydroxybutyrate, citrate, phenylalanine, 2-hydroxyisovalerate, and mannose, carboxymethylated, neuroketal, and malondialdehyde [78,83]. LC/MS/MS lipid profiling has correlated disease severity of multiple sclerosis with increased levels of 15-hydroxyeicosatetraenoic acid (15-HETE), prostaglandin E2, DHA-derived resolvin D1 (RvD1), and DHA-derived neuroprotectin D1 (NPD1) [84]. Surprisingly, similar perturbations were not observed in the serum samples from the same subjects. In these same studies, CSF and serum levels of thromboxanes, leukotrienes and prostaglandin D2 were comparable between multiple sclerosis and healthy controls. In a separate study, MRS coupled with neuroimaging discriminated multiple sclerosis from healthy controls by increased expression of glutamate, N-acetylaspartate (NAA), GABA, and aspartate [79]. 

Overall, perturbations in phospholipids, sphingolipids, amino acids, and long-chain polyunsaturated fatty acids appear to associate with multiple sclerosis pathology. It remains to be determined whether these differences reflect changes in myelin-specific lipid content as a consequence of demyelination, increased immune cell activity, or a combination of both. Further studies are needed to establish causal relationships and explore metabolic differences between female and male patients. It is intriguing that most metabolite perturbations correlated with immune cell survival, migration, and/or effector function. 

## 2. Metabolomic Profiling of NMOSD

### 2.1. Metabolomic Studies to Discriminate NMOSD 

Five untargeted and two targeted metabolomic studies have profiled, using multivariate and/or univariate analyses, metabolite perturbations in blood, plasma, CSF, or urine samples from NMOSD patients. These studies are listed here and highlight potential confounding factors, including metabolomic platform, origin of biopsy, state of disease at the time of biopsy collection, serology, biological sex, therapeutic pharmacological usage, and comparative control cohorts. Moussallieh et al. 2014 utilized proton High Resolution Magic Angle Spinning NMR spectroscopy (^1^H-HRMAS NMR) in order to compare sera from 44 NMO patients (22 AQP4-IgG^+^ and 22 AQP4-IgG^−^) with 47 RRMS patients and 42 healthy controls [85]; however, the reported results did not differentiate between AQP4-IgG^+^ and AQP4-IgG^−^ NMO. In a second, separate study, Gebregiworgis et al. 2016 used one-dimensional proton NMR spectroscopy (1D ^1^H-NMR) to compare urine samples from nine AQP4-IgG^+^ NMO patients undergoing remission, eight patients with RRMS, and 12 healthy controls [86]. In a third study, Park et al. 2016 applied GC-time of flight-mass spectrometry (GC-TOF-MS) to analyze CSF from 49 NMO patients (of unreported AQP4-Ab or MOG-Ab serology), 30 idiopathic transverse myelitis patients, 54 patients with RRMS, and 12 healthy controls [87]. In a fourth investigation, Kim et al. 2017 employed multiple forms of ^1^H-NMR spectroscopy, including one-dimensional proton nuclear overhauser effect spectroscopy (1D ^1^H-NOESY), ^1^H-^13^C heteronuclear single quantum coherence spectroscopy (^1^H-^13^C HSQC), and two-dimensional proton total correlated spectroscopy (2D ^1^H-TOCSY), in order to quantify and compare CSF metabolites from 57 NMO patients (of unreported AQP4-Ab or MOG-Ab serology) with 50 RRMS patients and 17 healthy controls [74]. Finally, Jurynczyk et al. 2017 applied a 1D ^1^H-NOESY pre-saturation scheme to ^1^H-NMR spectroscopy and subsequent lipoprotein profiling to compare plasma metabolites between 54 AQP4-Ab^+^ NMOSD, 20 MOG-Ab^+^ NMOSD, and 34 RRMS patients [82]. While Jurynczyk et al. was the only study to report the comparison of AQP4-Ab^+^ NMOSD and MOG-Ab^+^ NMOSD with RRMS, these investigations did not compare NMOSD or RMSS with healthy controls.

Although not explicitly self-identified as metabolomic investigations, at least two additional studies have incorporated targeted metabolomic platforms to evaluate NMOSD [88,89]. In the first of these two studies, Tortorella et al. 2011 used high performance liquid chromatography-mass spectrometry/more selective (HPLC-MS/MS) to quantify NAA from CSF and sera from 32 NMO patients (13 AQP4-Ab^+^ and 9 AQP4-Ab^−^), and compared these results with 48 RRMS and 76 healthy cohorts [88]. More recently, Cha et al. 2016 employed liquid chromatography-electrospray ionization tandem mass spectrometry with picolinyl ester derivatization (LC-ESI/MS/MS + PE) to quantify CSF 24S-, 25-, and 27-hydroxycholesterols (OHCs) in 26 AQP4-Ab^+^ NMO patients and 23 control patients with other non-inflammatory, non-degenerative neurological disorders (ONNDs) [89]. This publication further reported elevated serum OHC levels in NMOSD when compared to ONNDs, as detected by liquid chromatography-silver ion coordination ionspray tandem mass spectrometry (LC-Ag^+^CIS/MS/MS) [89]. Neither of these studies included healthy controls.

Collectively, these metabolomic investigaations identify significant perturbations of 36 metabolites in plasma, serum, blood, CSF, and urine biopsies collected from NMOSD patients as compared to RRMS and healthy controls (Table 1). Of these 36 metabolites, acetate, lysine, formate/formic acid, glucose, lactate/lactic acid, *N*-actyl aspartate (NAA), and scyllo-inositol discriminated NMOSD in two or more biopsy sources from two or more independent studies [74,82,85,86,87,88,89]. Of these eight metabolites, only one inter-study discordance was noted, and this was increased serum lysine [85] compared to decreased plasma lysine [82], in NMOSD compared to RRMS. A host of technical, but not known, biological possibilities could explain these lysine observations. In terms of technique, different sample sources, sample preparations, assay sensitivities, and metabolomic platforms for data analyses were used. Overall, the levels of a select few metabolites discriminate between NMOSD and multiple sclerosis and between AQP4-Ab^+^ NMOSD and MOG-Ab^+^. These include circulating levels of SCFA, lipoproteins, lipids, glycolysis intermediates, and essential amino acids, as illustrated in Figure 1. 

Six of the seven NMOSD metabolomic studies reviewed considered demographics, duration of disease, treatment, AQP4-Ab and MOG-Ab serology, age, biological sex, or MRI lesions in their data analyses [74,82,85,87,88,89]. Of these variables, increased CSF fatty acids or lactic acid levels [87], but decreased CSF isobutyrate levels discriminated relapsed and acute NMOSD [74]. Further, increased CSF OHC levels positively correlated with acute NMOSD disease severity, as determined by increased expanded disability status scale (EDSS) scores [89]. Otherwise, no metabolite perturbations correlated with these variables, but it is important to recognize that no study considered most of these confounding environmental factors. No study has yet addressed the sexual dimorphism associated with disease susceptibility to discriminate between AQP4-Ab^+^, MOG-IgG^+^, and seronegative NMOSD. 

Collectively, these data provide a supportive basis for the continued use of metabolic profiling for NMOSD biomarker discovery. Putative candidate diagnostic markers to differentiate NMOSD from RRMS include increased levels of lactate, alanine, unsaturated lipids, NAA, acetate, and lysine and decreased levels of histidine and glutamine (Figure 1). Putative biomarkers to discriminate between AQPA-Ab^+^ NMOSD and MOG-Ab^+^ NMOSD include increased levels of unsaturated fatty acids, concentration and size of large LDL particles, size of small LDL particles, glutamate, and glucose, formate, and leucine. To our knowledge, metabolic profiling has not been applied to AQPA-Ab^−^ MOG-Ab^−^ NMOSD. More studies are needed in order to validate these observations, and the consideration of clinical and environmental variables need to addressed in these future investigations. In the following paragraphs, we further discuss the relationship of these metabolite perturbations with respect to alterations in energy and fatty acid metabolism, immune cell function, and the gut bacterial microbiome.

### 2.2. NMOSD Metabolomics Profiling of SCFA in Fatty Acid Metabolism and Glycolysis 

Four independent studies support SCFAs as candidate discriminatory NMOSD biomarkers. The serum acetate levels were increased in NMOSD (22 AQP4-Ab^+^ and 22 AQP4-Ab^−^ patients) as compared to 47 RRMS and 42 healthy controls by 1.8- and 3.4-fold, respectively [85]. In agreement, a second study reported increased serum acetate levels in 54 AQP4-Ab^+^ NMOSD and 20 MOG-Ab^+^ NMOSD patients compared to 34 RRMS patients (82). This same study also showed that increased plasma formate levels discriminated the same 20 MOG-Ab^+^ NMOSD patients from the same 54 AQP4-Ab^+^ NMOSD and 34 RRMS patients. Further, increased urinary acetate, acetoacetate, and oxaloacetate levels were reported in AQP4-Ab^+^ NMOSD patients (*n* = 9) compared to healthy controls (*n* = 7) [86]. Increased plasma formate and leucine levels were shown to further discriminate 20 MOG-Ab^+^ NMOSD patients from 54 AQP4-Ab^+^ NMOSD and 34 RRMS patients [82]. In comparison, CSF acetate levels were decreased in 57 NMOSD patients, of unknown Ab serology, as compared to 17 healthy controls [74]. Discrimination between multiple sclerosis or NMOSD based on the 2010 McDonald and the 2015 International Panel for NMO diagnosis criteria leaves open the possibility that inclusion of NMOSD seronegative patients in the latter study as a possible reason for the discrepancy of NMOSD acetate compared to all other studies. 

Mechanistically, within the CNS, acetate is preferentially metabolized by astrocytes [74,86,87,90,91,92,93,94,95], and it plays a key role in energy bioavailability, production of myelin lipids [96,97], and lymphocyte effector function [98,99,100,101,102,103,104,105]. Commensal bacteria also produce acetate. Glutamate, which is the principal substrate for acetate synthesis, has also been implicated in NMOSD pathology [94]. AQP4-Ab internalizes EAAT-2 on astrocytes [106,107,108], which may alter acetate production. Formate is a monocarboxylic acid end product of choline and amino acid oxidation, and it is thought to be produced mostly within the mitochondria and released into the cytosol, where it plays a key role in one-carbon metabolism for purine synthesis, thymidylate synthesis, and methylation reactions. Formate is also a byproduct of cholesterol synthesis, bacterial metabolism, and environmental exposure [109]. In NMOSD, increased folate may contribute to glucose-independent one-carbon metabolism-mediated regulation of effector T cells [110]. Perhaps an overabundance of folate is a key factor driving clonal T cell proliferation in response to MOG-Ab? The role of leucine in NMOSD is also not clear; however, leucine is well recognized as a key regulator of the activation of immune responses [111].

Overall, these data support SCFA as potential NMOSD biomarkers. Further, the study of a possible gut-brain axis as a cause of NMOSD warrants further investigation given the recent observation of an overabundance of *C perfringens*, a bacterium linked to SCFA metabolism, in the gastrointestinal tract of NMOSD patients [112,113].

### 2.3. NMOSD Metabolomics Profiling of Lactate/Lactic Acid in Fatty Acid Metabolism and Glycolysis

Each of the five untargeted metabolomics studies reviewed here report the differences of lactic acid or its conjugate base, lactate, in NMOSD (Table 1). In particular, Moussallieh et al. reported that increased serum lactate levels discriminated 22 AQP4-Ab^+^ and 22 AQP4-Ab^−^ NMOSD patients from 47 RRMS and 22 healthy controls while using multivariate analysis [85]. Park et al. used univariate analysis to report that increased CSF lactic acid levels discriminated 57 NMOSD patients, of unknown Ab serology, from 50 RRMS patients and 17 healthy controls [74]. In support, Jurynczyk et al. 2017 further identified increased plasma lactate levels in a group of 54 AQP4-Ab^+^ NMOSD patients as compared to 34 RRMS patients following one-way ANOVA univariate analysis in order to generate a predictive OPLS-DA model that successfully discriminated AQP4-Ab^+^ NMOSD from RRMS with 92% accuracy [82]. Park et al. 2016 showed that increased CSF lactic acid and fumaric acid could discriminate 49 NMOSD patients (32 relapse and 17 remission, unknown Ab serology) from 54 RRMS patients, 30 idiopathic transverse myelitis (ITM) patients, and 12 healthy controls [87]. Park et al. 2016 further discriminated between NMOSD disease and relapse by the perturbation of a host of metabolites, including cyano-L-alanine, lactic acid, citric acid, homoserine, phenylalanine, myristic acid, and salicyladehyde. Lastly, Gebregiworgis et al. 2016 reported that reduced urinary lactate and several other urinary metabolite perturbations differentiated nine AQP4-Ab^+^ NMOSD patients from seven healthy controls. 

Overall, increased levels of lactate are consistent with a role for dysregulated fatty acid metabolism in NMOSD pathology. The uniformity of results between serum and CSF further increase the likelihood of lactate as a candidate biomarker for NMOSD diagnosis as well as disease relapse. Mechanistically, astrocytes, in comparison to neurons, preferentially use glycolysis and fermentation for energy metabolism, leading to lactate accumulation, the formation of the astrocyte-neuron lactate shuttle (ANLS), and enhanced neuroexcitation, and the activation of immune responses [114,115,116,117,118,119,120]. Remarkably, prior in vitro studies have suggested that lactic acid upregulates AQP4 expression in astrocytes and promotes the production pro-inflammatory cytokines IL-17 and IL-23 [121,122,123,124]. 

### 2.4. NMOSD Metabolomics Profiling of Lipids and Lipoproteins in Energy Metabolism and Glycolysis

The CNS is highly cholesterol-rich. The majority of cholesterol is found in the myelin sheaths. A lesser amount is present in glial and neuronal cell plasma membranes. Lipoproteins contain lipids and proteins, and transport lipids. CNS cholesterol is mostly derived by de novo synthesis, as the BBB prevents the uptake of lipoprotein cholesterol from the circulation. The lipoprotein composition in the CNS differs from that in the circulation, and neurons and astrocytes coordinate lipoprotein metabolism in the brain. 

Plasma lipoproteins, along with perturbations of amino acids, scyllo-inositol, and myo-inositol, have been shown to discriminate between AQP4-Ab^+^ NMOSD, AQP4-Ab^−^ NMOSD, and multiple sclerosis. Standard NMR is not equipped to categorize individual lipoprotein subpopulations or measure lipoprotein particle number or size. Thus, Jurynczyk et al. first applied an NMR-based lipidomics platform to plasma samples, and then applied OPLS-DA to discriminate 54 AQP4-Ab^+^ NMOSD from 20 AQP4-Ab^−^ NMOSD and 34 RRMS by increased levels of large HDL particles and glucose, and decreased levels of small HDL particles, phosphocholine/lipoprotein, and scyllo-inositol [82]. Increased large LDL particles levels discriminated AQP4-Ab^+^ NMOSD and RRMS patients from AQP4-Ab^−^ NMOSD. Increased plasma levels of formate and leucine, along with decreased plasma myo-inositol levels, discriminated AQP4-Ab^−^ NMOSD from AQP4-Ab^+^ NMOSD and RRMS patients. Additionally, RRMS differentiated from AQP4-Ab^+^ NMOSD from AQP4-Ab^−^ NMOSD by increased plasma levels of histidine, lysine, creatinine, and creatine, but decreased large HDL particles, lactate, and alanine. The total number of HDL particles, the total number of LDL particles, and the level of total HDL, LDL, and triglyceride levels were not discriminatory between these three patient groups [82].

A second, independent study used enzyme-linked immunosorbent assay (ELISA), 12-hour fasting blood draws, clinical lipid tests, and covariance analysis with age and gender as covariants in 56 NMOSD patients, 53 RRMS patients, and 54 healthy controls [125]. In this study, Li et al. discriminated NMOSD (females combined with males) from RRMS (females combined with males) by increased serum apolipoprotein B (ApoB) level and ratio of ApoB to ApoA1. The female NMOSD patients differentiated from male NMOSD patients by decreased serum total cholesterol and LDL levels. Serum total cholesterol, LDL, ApoA1, and total cholesterol/HDL-cholesterol did not differ between NMOSD and RRMS patients.

A third report by Cha et al. highlighted a role for hydroxycholesterols in NMOSD pathology and suggests oxysterols as candidate biomarkers for NMOSD diagnosis [89]. Serum and CSF samples from 26 AQP4-Ab^+^ NMO patients and 23 patients with other non-inflammatory, non-degenerative neurological disorders (ONND) by LC-MS/MS. AQP4-Ab^+^ NMO patients were differentiated from ONND patients by increased CSF, as well as serum level of 25-hydroxycholesterol (OHC), 27-OHC, and the ratio of CSF 27-OHC to 24S-OHC, which could be interpreted as either increased CNS synthesis of 27-OHC or increased the permeability of the BBB. CSF 24-OHC did not associate with disease activity; however, it did associate with the number of CNS inflammatory cells. Thus, CSF 24-OHC levels may increase as a result of immune-mediated CNS injury. Further, the serum and CSF levels of 24S-OHC, 245-OHC, and 27-OHC were correlated with NMOSD disease severity (EDSS) at acute attack. Only CSF 27-OHC correlated with disability (*r* = 0.521, *p* = 0.009). It is not yet clear why CSF 27-OHC is increased and associated with disease disability in NMOSD. However, 27-OHC is mostly synthesized from cholesterol by cholesterol 27-hydroxylase (CYP27), thus perhaps astrocyte and glial cell damage alters de novo synthesis of cholesterol in the CNS in NMOSD. The levels of CSF 24S-OHC, serum 24S-OHC, serum 25-OHC, and serum 27-OHC did not differ between NMOSD and ONND. It is speculative that dietary intake contributes to the changes in serum total cholesterol and serum LDL levels that are observed between female and male NMOSD patients, but more studies are needed to explore this further.

Overall, these data collectively provide a strong foundation to implement CNS-restricted and circulating levels of lipids and lipoproteins as strong candidate biomarkers to discriminate between AQPA4-Ab^+^ NMOSD and AQP4-Ab^−^ NMOSD, between NMOSD and related autoimmune diseases, and between NMOSD and other CNS disorders that are neither inflammatory nor neurogenerative, such as polyneuropathy, cranial nerve palsy, radiculopathy, and spinal arteriovenous malformation. It is likely that a more complete understanding of the role(s) of lipids and associated lipoproteins in NMOSD pathology will enhance our knowledge of why females are more disease susceptible than are males. Recently, 25-OHC has been shown to stimulate toll-like receptors on immune cells [126]. In this regard, cholesterol and its metabolites join eicosanoids and sphingosine-1-phosphate as important modulators of immune responses to environmental stimuli.

### 2.5. NMOSD Metabolomic Profiling of Amino Acids in NMOSD

Amino acids are well recognized as a key energy source, a substrate for protein synthesis, and a regulator of rapamycin (mTOR), a central regulator of innate and adaptive immune cells responses to the local environment. A select panel of amino acids have been shown to discriminate between AQP4-Ab^+^ NMOSD, MOG-Ab disease, and RRMS, as diagrammed in Figure 2. These include increased serum glutamate, lysine, *N*-acetylaspartate (NAA), and leucine, but decreased serum histidine and glutamine. Increased circulating levels of glutamate discriminate AQP4-Ab^+^ NMOSD from MOG-Ab^+^ NMOSD and RRMS. Glutamate is the most abundant amino acid in the CNS and the major CNS excitatory neurotransmitter. When in excess, it also contributes to neuronal dysfunction and destruction. Excess CNS glutamate is consistent with a loss of both astrocytes and neurons, the two major cell types that express high-affinity glutamate uptake transporters. Neuronal mitochondria synthesize NAA. Dysregulated NAA is suggestive of mitochondrial dysfunction and often results in cell death. Neither the source nor the consequence of decreased histamine in NMOSD compared to RRMS is well understood. Taken together, the amino acid profile that discriminate between AQP4-Ab^+^ NMOSD, MOG-Ab^+^ NMOSD, and RRMS appears to have promise for promise for new discriminatory biomarker discovery. Yet, further investigations are needed for the validation and consideration of other related overlapping neurological and autoimmune disorders.

### 2.6. Magnetic Resonance Spectroscopy (MRS) in NMOSD Metabolomic Studies

Magnetic Resonance Spectroscopy (MRS) is a platform that enables noninvasive evaluation of metabolites in vivo, and MRS Imaging (MRSI) further adds a dimension of spatial localization to this technology. MRS and MRSI have both demonstrated feasibility of quantifying lipids, amino acids, lactate, choline, NAA, creatine, and myo-inositol in CNS tissues of patients with neurological disorders [127]. At least five MRS studies have investigated in vivo metabolite alterations in NMOSD, and only one reported significant metabolite perturbations [128,129,130,131,132]. This single study showed increased choline and NAA in the normal appearing white matter (NAWM) and normal appearing gray matter (NAGM) of NMOSD patient’s brains compared to RRMS patients and healthy controls [129]. This finding is in agreement with a separate study showing equivalent or reduced serum or CSF levels of NAA in NMOSD as compared to RRMS and healthy controls [88]. New technological innovations continue to emerge. For instance, high spatial resolution matrix-assisted laser desorption ionization (MALDI) imaging MS evaluates optic nerve anatomy as well as lipid and protein profiles [133]. These new developments further promote advances in a precision medicine approach to treat NMOSD.

## 3. Proteomics in NMOSD 

### 3.1. Proteomic Studies to Discriminate NMOSD

Appendix A presents a summary of these data. Beyond metabolomic evaluations of small metabolites, at least eight studies have used MS or NMR platforms to evaluate proteomics in NMOSD [134,135,136,137,138,139,140,141] Of these, six studies quantified and compared NMOSD spectral profiles to reference groups [134,135,136,137,138,139]. Lee et al. 2016 used high resolution hybrid LTQ-orbitrap MS to profile 442 CSF exosome proteins from 10 NMOSD, 12 LETM, and 10 RMMS patients. This study identified significant perturbation of 123 proteins in NMOSD with respect to LETM and RMMS patients, most notably increased levels of glial fibrillary acidic protein (GFAP) [134]. Jiang et al. used HD-MS/MS or 2-DE and HD-MS/MS to profile CSF or serum protein perturbations in six NMOSD patients compared with six RRMS patients and six healthy controls [135,136]. Bai et al. utilized 2-DE and MALDI-TOF-MS to profile CSF proteomics of NMOSD patients and patients other neurological disorders (OND), including tension-type headaches, drug-induced delirium, normal pressure hydrocephalus, and trigeminal neuralgia [137,138]. Lastly, Nielsen et al. used LC-MS/MS to quantify urinary protein concentrations from 32 AQP4-Ab^+^ NMOSD patients, 46 RMSS patients, and 31 healthy controls [139]. Overall, these six studies identified the concentrations of 162 protein perturbations in NMOSD with respect to RRMS, LETM, OND, and/or healthy controls [134,135,136,137,138,139]. Two additional studies did not focus on protein identification, but rather used MALDI-TOF-MS to discriminate CSF spectral peaks between NMOSD and RRMS [140] or LC-ESI/MS/MS to combine CSF transcriptome and proteome profiling in NMOSD patients [141]. 

Collectively, of the 162 protein perturbations that were identified in NMOSD, haptoglobin (Hp), immunoglobulin kappa (Igκ) and Ig lambda-2 chain C regions (IGLC2), neurofilament, apolipoproteins, collagen alpha-1 chains, contactin-1, keratin proteins, pigment-epithelium derived factor (PEDF), and transthyretin (TTR) differed (*p* < 0.05) in two or more independent studies and, with the exception of neurofilament, by two independent laboratories [134,135,137,138]. In addition, haptoglobin, IGLC2, collagen alpha-1 chains, contactin-1, and keratin protein perturbations were detected in more than one biofluid from NMOSD patients [134,135,137,138]. 

### 3.2. Proteomic-Based Acute Phase Protein Perturbations

Appendix A presents a summary of these data. Haptoglobin (Hp) is a standout endpoint in NMOSD proteomics, as it has the highest number of studies supporting its increase in NMOSD with respect to control cohorts and has a high magnitude of increase [134,135,137,138]. Most notably, a two- to three-fold increase in serum haptoglobin allelic subtype two (Hp2) has been reported in NMOSD with respect to RRMS and healthy controls [135]. Conversely, ELISA-based studies reported equivocal levels of serum Hp amongst NMOSD, RRMS, Alzheimer’s disease (AD), and healthy controls [135,142]. The rationale underlying increased levels of Hp in NMOSD is not clear. However, Hp, which is best known for its antioxidant activity [143,144,145,146,147], is produced in abundance by oligodendrocytes and astrocytes, and it is known to modulate monocyte, granulocyte, neutrophil, and lymphocyte cell activity [148,149,150,151,152,153,154]. As such, the role of Hp and Hp2 in AQP4-reactive B and T cells is of particular interest due to their ability to affect neuron function and immune responses. A number of other acute phase proteins also follow a trend of predicted inflammation in NMOSD [155]. For example, CSF levels of ceruloplasmin, alpha-2-macroglobulin, and fibrinogen are increased in NMOSD compared to LETM. The CSF levels of retinol-binding protein 4 and TTR are decreased in NMOSD compared to LETM. TTR, a marker of impaired BBB integrity, in combination with Hp, is a promising biomarker for diagnosis Parkinson’s Disease [156,157]. Future studies are needed in order to evaluate a causal role for Hp and/or TTR in NMOSD disease and the potential of these two acute phase proteins for disease discrimination of NMOSD patients.

### 3.3. Proteomic-Based Perturbations in Humoral Immunity

Appendix A present a summary of these data. Deposits of anti-AQP4-Ab, anti-MOG-Ab, and complement components are characteristic of active NMO lesions [158]. Increased AQP1-IgG, Ig gamma–3 (IgG3), Igκ, and Ig lamda (Igλ) levels are also observed in NMOSD as compared to LETM, RRMS, OND, and healthy controls [43,134,137,139,159,160]. These increased Ab-related proteins may correlate with circulating targeted Abs against AQP4, MOG, and perhaps other protein epitopes, such as glucose-regulated protein 78, Kir 4.1 K^+^ channel, and C1q [161,162,163,164,165,166,167]. The concentrations of at least 15 different complement-related components, including C4b-binding and Hp-related proteins, are reported increased in NMOSD compared to LETM. The classical complement pathway has been identified as the third-most significant proteomic pathway associated with NMOSD [134]. ELISA platforms further corroborate complement components as possible biomarkers to discriminate NMOSD from multiple sclerosis [166,168,169,170,171,172]. 

### 3.4. Targeted Immunoassay-Based Studies to Discriminate NMOSD

Appendix A presents a summary of these data. In contrast to metabolomics and proteomics, targeted immunoassays have been used for decades and remain, in some cases, a preferred platform for quantifying targeted proteins. More than 55 different bead-based, ELISA, or other targeted immunoassay platforms have been applied to quantify circulating levels of cytokines, chemokines, inflammatory mediators, complement, cell damage products, or other proteins in NMOSD. A total of 119 serum protein changes have been reported to differentiate between NMOSD and RRMS, OND, or healthy controls. As many of these perturbations have previously reviewed [173,174,175], here our discussion will focus on the perturbations that have not previously been reviewed.

A minimum of36 independent studies have quantified the concentrations of 86 unique CSF analytes from NMOSD patients [134,137,142,164,170,172,176,177,178,179,180,181,182,183,184,185,186,187,188,189,190,191,192,193,194,195,196,197,198,199,200,201,202,203,204,205]. Several of these studies corroborate the metabolomic and proteomic observations discussed above. For example, targeted immunoassays support increased levels of Hp [137,142] and GFAP [179,186,188,196,199,200] in NMOSD with respect to all comparison cohorts studied, with the exception of acute disseminated encephalomyelitis (ADEM), by as much as 1000-fold. In addition, targeted immunoassays have been used to recognize lactate as a candidate biomarker to discriminate between NMOSD relapse and NMOSD remission [43]. Moreover, ELISA is a gold standard for measuring secreted protein and has been successful in confirming increased levels of proinflammatory IL-17, a potent neutrophil chemotactic cytokine, in NMOSD pathology [181,184,201]. Likewise, ELISA assays have substantiated increased levels of IL-6, a cytokine that suppresses T_REG_ differentiation and induces B cell differentiation, in NMOSD [137,164,176,177,179,180,181,184,188,189,190,195,200,201,204]. The CSF levels of IL-2 and interferon-γ (IFN-γ), cytokines associated lymphocyte effector function, appear to remain unchanged in NMOSD when compared to controls [134,176,179,181,184,190,195]. Fewer studies have investigated the CSF levels of TGF-β in NMOSD, and the results collected to date are inconclusive [179]. 

Regarding serum, at least 28 independent targeted immunoassay-based studies analyzed serum from NMOSD patients [142,164,169,170,171,179,183,185,187,190,191,192,194,200,203,205,206,207,208,209,210,211,212,213,214,215,216,217]. These results of these studies are summarized in Appendix A. Overall, these results showed increased levels of IL-17A and IL-17F in NMOSD compared to RRMS, [190,206,207,208,212,214]. However, the levels of IL-6 [179,190,207,208,213,214] and IFN-γ [210,211,213,214] in NMOSD compared to RRMS patients are somewhat controversial.

In regards to plasma, at least six independent studies have used targeted immunoassay platforms to evaluate the levels of a combination of 24 proteins in NMOSD patients when compared to RRMS patients and healthy controls [166,168,206,218,219,220]. Appendix A summarizes the cytokine e.g., IL-1β, IL-6, IL-17, and TNF-α, and complement components, e.g., C1s, C3a, C4a, C4d, and C5, which are consistently elevated in NMOSD as compared to RRMS and healthy controls. 

Whole blood sample investigations have identified at least 12 proteins that differ between NMOSD patients as compared other overlapping related disorders and healthy controls (Appendix A). Collectively, these studies have identified that increased levels of BAFF receptor, C-X-C motif chemokine receptor 5 (CXCR5), IFN-γ, IL-6, and IL-12, but decreased levels of IL-10 levels in NMOSD when compared to RRMS, SPMS, other inflammatory neurological disorders (OIND), and healthy controls [164,177,202]. It remains to be determined why the particular profile of inflammatory markers differs between NMOSD and other inflammatory diseases.

Taken together, gold standard targeted immunoassays support the observation that NMOSD discriminates from other related overlapping inflammatory disorder with respect to CNS-restricted and circulating levels of cytokines, chemokines, and other soluble mediators that relate to innate and adaptive immune responses, demyelination, astrocyte destruction, neuronal dysfunction, and myelin and plasma membrane repair. 

## 4. Conclusions and Future Perspectives 

The undertaking of metabolomic investigations to establish discriminative biomarkers for NMOSD has begun and initial studies show great promise, as they associate a battery of metabolites that link cells, proteins, metabolic pathways, and soluble mediators that may facilitate the pathophysiology of NMOSD. A select group of SCFAs, lipids, lipoproteins, and amino acids that are associated with energy metabolism and glycolysis appears to be particularly promising as candidate biomarkers for disease diagnosis, progression, and response to treatment. Yet, several limitations are apparent, and they identify the need for additional studies to advance NMOSD biomarker discovery. These include, but are not restricted to, the standardization of biofluids and sample preparation for analyses. There is no doubt that the biology underlying NMOSD has rapidly advanced over the past decade. Given the importance of early disease detection and the adverse effects to treatment in response of misdiagnosis, there is a great need to classify NMOSD subtypes, at minimum, AQP4- and MOG-Ab serology, prior to sample analyses. There is also a tremendous need to predict NMOSD relapse, thus also highlighting the need to discriminate between samples collecting from patients during disease onset, disease remission, and disease relapse. One of the most puzzling unanswered questions is why is the intendent of NMOSD greater in females as compared to males. The ratio worldwide is currently estimated at 8–10 females to one male. A better understanding of this sex dimorphism is likely to add tremendous value in only novel biomarker discovery, but the likelihood of finding a cure as well. The identification for a role in disease pathology by environmental factors is now emerging. Thus, a possible role of a gut-brain axis should not be ignored in future study designs. In summary, metabolomic profiling of NMOSD has successfully identified a host of candidate biomarkers for NMOSD diagnosis, but more research is needed. 

## Figures and Tables

**Figure 1 metabolites-10-00374-f001:**
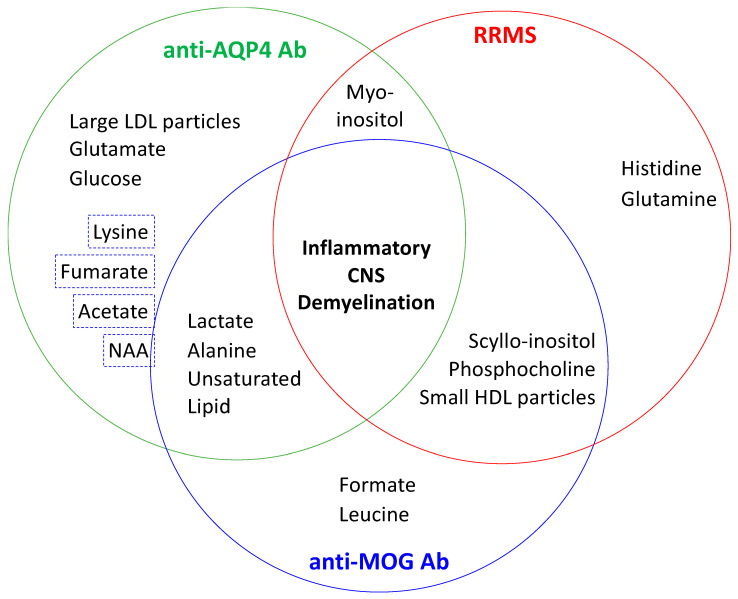
Circulating Metabolites that Discriminate between AQP4-Ab^+^ NMOSD, MOG-Ab^+^ NMOSD, and RRMS. The clinical manifestations of auaporin-4 (AQP4)-antibody (Ab) neuromyelitis optica spectrum disorder (NMOSD), myelin oligodendrocyte glycoprotein (MOG)-Ab NMOSD, and relapsing remitting multiple sclerosis (RRMS) overlap. Given that (1) the clinical features of NMOSD overlap with other central nervous system (CNS) demyelinating inflammatory disorders, (2) a subtype of AQP4-Ab^−^ and MOG-Ab^−^ NMOSD has been proposed, (3) circulating Abs may not be detectable following treatment or during disease remission, and (4) circulating levels of Abs are not prognostic—Ab-independent biomarkers to accurately discriminate NMOSD from other CNS disorders as well as between NMOSD subtypes is needed. Shown are the proposed circulating metabolites, for which varying levels of evidence exist, that discriminate or overlap between AQP4-Ab^+^ NMOSD, MOG-Ab^+^ NMOSD, and RRMS. Each metabolite indicated is increased relative to other(s) disease. Not enough evidence is yet available to propose whether fumaric acid, acetate, and/or N-acetylaspartate (NAA) discriminates AQP4-Ab^+^ NMOSD from MOG-Ab^+^ NMOSD and RRMS or discriminates AQP4-Ab^+^ NMOSD and MOG-Ab^+^ NMOSD from RRMS. The essential amino acid lysine, a precursor of glutamate in the CNS, is an interesting possibility that is supported by inconsistent published results that may be resolved with future studies that account for the state of disease, i.e., relapse or remission, at the time of sample collection.

**Figure 2 metabolites-10-00374-f002:**
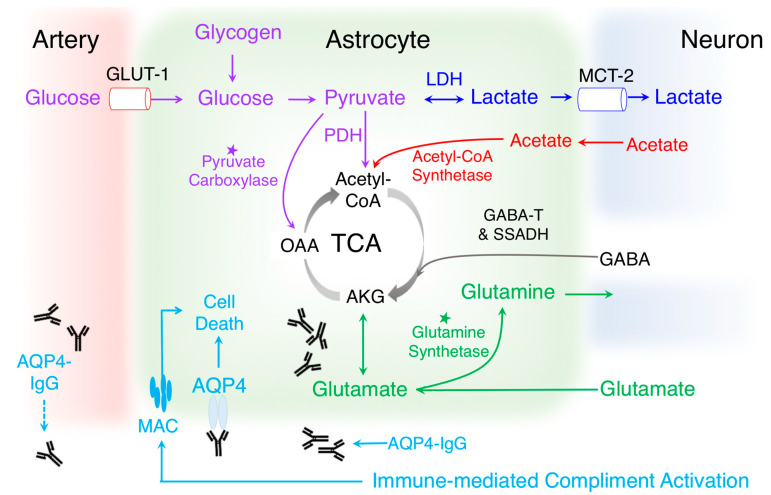
A role for metabolism in NMOSD. The role of astrocytes in neuronal-glial energy metabolism (red), the astrocyte-neuron lactate shuttle (ANLS) (blue), and the glutamate/glutamine shuttle system (green). The pathology of NMOSD, at least in most patients, has been traced to AQP4-Ab, which leads to complement- and antibody-mediated injury and death of astrocytes and neurons. This antigen-specific humoral response has direct implications on astrocyte-regulated neuron-glial energy metabolism and the neurotransmitter production as the glutamate/glutamine shuttle is dependent on the astrocyte-specific glutamine synthetase. These metabolic changes further impact the production of lactate by astrocytes. Abbreviations: AKG, alpha-ketoglutarate; acetyl-CoA; AQP4-Ab, aquaporin-4 autoantibody; GABA, gamma-aminobutyric acid; GABA-T, GABA-transaminase; GDH, glutamate dehydrogenase; LDH, lactate dehydrogenase; MAC, membrane attack complex; OAA, oxaloacetate; PDH, pyruvate dehydrogenase; SSADH, succinic semialdehyde dehydrogenase. * Astrocyte-compartmentalized enzyme.

**Table 1 metabolites-10-00374-t001:** Altered Metabolite Outcomes in Neuromyelitis Optica Spectrum Disorder (NMOSD) Metabolomic Studies.

Endpoint	Change	Biopsy	NMO Patients		Disease State ^a^		AQP4-Ab		Control ^b^	Method ^c^	Ref.
			Female	Male	Relapse	Remission	+	−			
1-Monostearin	↑	CSF	43	6	32	17	NR	NR	12 HC	GC-TOF-MS	[87]
1-Monopalmitin	↑	CSF	43	6	32	17	NR	NR	12 HC	GC-TOF-MS	[87]
2-Hydroxybutyrate	↑	CSF	51	6	36	21	NR	NR	17 HC	1D & 2D ^1^H-NMR	[74]
3-Hydroxybutyrate	↓	Urine	6	3	⎯	9	9	⎯	8 RRMS, 7 HC	1D ^1^H-NMR	[86]
3-Hydroxyisovalerate	↓	Urine	6	3	⎯	9	9	⎯	8 RRMS	1D ^1^H-NMR	[86]
3-Hydroxypropionic acid	↓	CSF	43	6	32	17	NR	NR	12 HC	GC-TOF-MS	[87]
25-Hydroxycholesterol	↑	CSF	19	7	26	⎯	26	⎯	23 ONND	LC-ESI/MS/MS + PE	[89]
27-Hydroxycholesterol	↑	CSF	19	7	26	⎯	26	⎯	23 ONND	LC-ESI/MS/MS + PE	[89]
Acetate	↑	Serum	32	12	NR	NR	22	22	47 RRMS, 42 HC	^1^H HRMAS NMR	[85]
	↓	CSF	51	6	36	21	NR	NR	17 HC	1D & 2D ^1^H-NMR	[74]
Acetone	↑	CSF	51	6	36	21	NR	NR	17 HC	1D & 2D^1^H-NMR	[74]
Alanine	↑	Plasma	46	8	NR	NR	56	⎯	34 RRMS + 20 MOG^+^	1D ^1^H-NOESY	[82]
Butane-2,3-Diol	↓	CSF	43	6	32	17	NR	NR	12 HC	GC-TOF-MS	[87]
Citrate	↓	CSF	51	6	36	21	NR	NR	50 MS	1D & 2D ^1^H-NMR	[74]
Creatinine	↓	Urine	6	3	⎯	9	9	⎯	8 RRMS, 7 HC	1D ^1^H-NMR	[86]
Formate	↑	CSF	51	6	36	21	NR	NR	17 HC	1D & 2D ^1^H-NMR	[74]
(MOG-Ab^+^) *	↑	Plasma	46	8	NR	NR	56	⎯	34 RRMS + AQPA-Ab^+^	1D ^1^H-NOESY	[82]
Fumaric Acid	↑	CSF	43	6	32	17	NR	NR	12 HC, 54 RRMS	GC-TOF-MS	[87]
Glucose	↓	CSF	51	6	36	21	NR	NR	17 HC	1D & 2D ^1^H-NMR	[74]
	↑	Plasma	46	8	NR	NR	56	⎯	34 RRMS + 20 MOG^+^	1D ^1^H-NOESY	[82]
Glutamate	↑	Serum	32	12	NR	NR	22	22	47 RRMS, 42 HC	^1^H HRMAS NMR	[85]
Glutamine	↓	Serum	32	12	NR	NR	22	22	47 RRMS, 42 HC	^1^H HRMAS NMR	[85]
Histidine	↓	Plasma	46	8	NR	NR	56	⎯	34 RRMS + 20 MOG^+^	1D ^1^H-NOESY	[82]
Inosine	↓	CSF	43	6	32	17	NR	NR	12 HC	GC-TOF-MS	[87]
Lactate/Lactic Acid	↑	CSF	51	6	36	21	NR	NR	50 MS, 17 HC	1D & 2D ^1^H NMR	[74]
	↑	Plasma	46	8	NR	NR	56	⎯	34 RRMS	1D ^1^H-NOESY	[82]
	↑	Serum	32	12	NR	NR	22	22	47 RRMS, 42 HC	^1^H HRMAS NMR	[85]
	↑	CSF	43	6	32	17	NR	NR	12 HC, 54 RRMS	GC-TOF-MS	[87]
	↓	Urine	6	3	⎯	9	9	⎯	8 RRMS, 7 HC	1D ^1^H-NMR	[86]
Large LDL particles, Concentration	↑	Plasma	46	8	NR	NR	56	⎯	34 RRMS + 20 MOG^+^	1D ^1^H-NOESY	[82]
Large LDL particles, Size	↑	Plasma	46	8	NR	NR	56	⎯	34 RRMS + 20 MOG^+^	1D ^1^H-NOESY	[82]
Leucine (MOG-Ab^+^) *	↑	Plasma	46	8	NR	NR	56	⎯	34 RRMS + AQPA-Ab^+^	1D ^1^H-NOESY	[82]
Lysine	↓	Plasma	46	8	NR	NR	56	⎯	34 RRMS	1D ^1^H-NOESY	[82]
	↑	Serum	32	12	NR	NR	22	22	47 RRMS	^1^H HRMAS NMR	[85]
	↓	Serum	32	12	NR	NR	22	22	42 HC	^1^H HRMAS NMR	[85]
Methylmalonate	↓	Urine	6	3	⎯	9	9	⎯	8 RRMS, 7 HC	1D ^1^H-NMR	[86]
Myo-Inositol (MOG-Ab^+^) *	↓	Plasma	46	8	NR	NR	56	⎯	34 RRMS	1D ^1^H-NOESY	[82]
NAA	↓	CSF	24	8	10	22	13	19	48 RRMS	HPLC-MS/MS	[88]
	↓	Serum	24	8	10	22	13	19	48 RRMS	HPLC-MS/MS	[88]
Oxaloacetate	↑	Urine	6	3	⎯	9	9	⎯	7 HC	1D ^1^H-NMR	[86]
Phosphocholine/lipoprotein Lipoprotein	↓	Plasma	46	8	NR	NR	56	⎯	34 RRMS + 20 MOG^+^	1D ^1^H-NOESY	[82]
Pyroglutamate	↑	CSF	51	6	36	21	NR	NR	17 HC	1D & 2D ^1^H-NMR	[74]
Salicylaldehyde	↑	CSF	43	6	32	17	NR	NR	12 HC	GC-TOF-MS	[87]
Scyllo-Inositol	↓	Serum	32	12	NR	NR	22	22	47 RRMS	^1^H HRMAS NMR	[85]
	↓	Plasma	46	8	NR	NR	56	⎯	34 RRMS + 20 MOG^+^	1D ^1^H-NOESY	[82]
Small HDL Particles	↓	Plasma	46	8	NR	NR	56	⎯	34 RRMS + 20 MOG^+^	1D ^1^H-NOESY	[82]
Threose	↓	CSF	43	6	32	17	NR	NR	12 HC	GC-TOF-MS	[87]
Unsaturated Lipid	↑	Plasma	46	8	NR	NR	56	⎯	34 RRMS	1D ^1^H-NOESY	[82]

1D ^1^H-NMR, one dimensional proton nuclear magnetic resonance spectroscopy; 1D ^1^H-NOESY, 1D ^1^H overhauser effect spectroscopy; 1D & 2D ^1^H-NMR, 1D & two dimensional ^1^H-NMR; AQP4-Ab, aquaporin-4 autoantibody; CSF, cerebrospinal fluid; HC, healthy controls; HDL, high density lipoprotein; ^1^H HRMAS NMR, ^1^H high resolution magic angle spinning NMR; HPLC-MS/MS, high performance liquid chromatography-mass spectrometry/more selective; GS-TOF-MS, gas chromatography time-of-flight mass spectrometry; LC-ESI-MS/MS + PE, liquid chromatography-electrospray ionization-tandem mass spectrometry with picolinyl ester derivatization; LDL, low density lipoprotein; MOG, myelin oligodendrocyte glycoprotein; MS, multiple sclerosis; NAA, *N*-acetyl aspartate; NMOSD, neuromyelitis optica spectrum disorder; NR, not reported; ONND, other noninflammatory neurological disorders; RRMS, relapsing-remitting MS. ^a^ NMOSD patient disease status at the time of biopsy sampling, ^b^ Statistical significance reported between NMOSD and each comparison group, ^c^ Method of metabolomic spectral data acquisition. * Indicates that MOG-Ab^+^ biopsies differed from AQPa-Ab^+^ and RRMS biopsies.

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
