# Peer review of "Metabolomic Profiling in Neuromyelitis Optica Spectrum Disorder Biomarker Discovery"

_metabolites, 2020, doi:10.3390/metabo10090374_

Round 1

Reviewer 1 Report

Dear Authors,

my comments are here listed:

1) At page 3, authors must cite papers of Pisani F. et al., relative to the the Anti-AQP4 detection in NMO sera and in particular the AQP4-isoform dependent  sensitivity of the cell-based assay (CBA) (PMID: 24260168).

2) The sentence "In turn, pathogenic AQP4-Ab binds either OAP assemblies or M1 and/or M23 isoforms on astrocyte endfeet, initiating astrocyte lysis and complement-dependent oligodendrocyte injury" must be changed because it is generally accepted that AQP4-IgGs are abl to bind exclusively the OAP at level of extracellular and conformational epitopes. To support this author must cite (PMID: 21212277).

3) In the paragraph 1.5 Author must consider that AQP4-IgG seronegative may be caused by the use of a low sensitivity assay. In my hands many case of "AQP4-IgG" negative false-negative were simply for the use of a low sensitivity method.

General Comment:

Desite, actually, the use of Metabolomics and proteomics is far to be a simple and high performance-test to support the NMOSD diagnosis, i believe that this field of study must be apprecciate and developed, expecially for new biomarker discovery. For these reasons comprensive review are always useful to increase
our understanding of the NMOSD.

Author Response

1) At page 3, authors must cite papers of Pisani F. et al., relative to the the Anti-AQP4 detection in NMO sera and in particular the AQP4-isoform dependent  sensitivity of the cell-based assay (CBA) (PMID: 24260168). We have inserted this reference as recommended - Thank you for your suggestion!

2) The sentence "In turn, pathogenic AQP4-Ab binds either OAP assemblies or M1 and/or M23 isoforms on astrocyte endfeet, initiating astrocyte lysis and complement-dependent oligodendrocyte injury" must be changed because it is generally accepted that AQP4-IgGs are abl to bind exclusively the OAP at level of extracellular and conformational epitopes. To support this author must cite (PMID: 21212277). We have made the changes as suggested - Thank you for your comments.

3) In the paragraph 1.5 Author must consider that AQP4-IgG seronegative may be caused by the use of a low sensitivity assay. In my hands many case of "AQP4-IgG" negative false-negative were simply for the use of a low sensitivity method. We have made the change as suggested - Thank you for the comment.

General Comment:

Desite, actually, the use of Metabolomics and proteomics is far to be a simple and high performance-test to support the NMOSD diagnosis, i believe that this field of study must be apprecciate and developed, expecially for new biomarker discovery. For these reasons comprensive review are always useful to increase
our understanding of the NMOSD. We agree with your comments.

Reviewer 2 Report

Thomas and McKarns did an extensive review analysing the metabolomic perturbations and proteomics outcomes that may discriminate between Neuromyelitis Optica Spectrum Disorder disease (NMOSD) from demyelinating inflammatory disorders as MS, among others. They are trying to stablish a “definitive metabolomic signature” able to be applied to improve diagnosis, treatment, and prognosis of NMOSD.

The literature analysis was well executed, profound and the “conclusions” (see later comment about that) are consistent with provided data. However, some comments would be done to improve the clearness of the manuscript.

The manuscript is sometimes difficult to be follow in a grammatical point of view. The literature data are exposed very “telegraphically” with one after other reference without, at least for me, a clear line of thinking. Let me refer some examples:

1) Section 1.5 (pg. 3). The use of commas (,) and point and commas (;) to list changes in NMOSD disease is confusing and not “well accepted” as clearest and grammatical use. This way of writing is followed along the manuscript several times.

2) Section 1.6.1 (pg. 4). I believe that for a journal named “Metabolites” is a bit un-sense to give definitions as “metabolism”, “metabolites” or “metabolomics”. May be the overview should be centred (as suggestion) in quality of data provided by metabolomics in comparison with other techniques. Of course, it is a prerogative of the author to decide how to write the manuscript.

3) In several places of the manuscript are used too “colloquial words or sentences” or “phrase connectors”. For instance, (pg. 8, 2nd paragraph) “With this in mind, six…”, (pg. 8, 3rd paragraph) “…NMOSD. Again, the specificity…” (pg. 9, 3rd paragraph), “…metabolism. Further, given that…”, (pg. 13, 3rd paragraph) “…activity (198-204). As such, due to their…”, (pg. 15, 1st paragraph) “…To begin this exhilarating journey…”, (pg. 16, 1st paragraph) ”…of novel discriminative biomarkers and therapeutic strategies are great”.

There are also some mistakes that should be corrected:

1) The “supplementary Table 1” or “Table 1” is written in bold letter in some cases and in plain letter in others.

2) The “supplementary Table 3” is referred in the text BEFORE that “supplementary Table 2”. Then, the message of those tables in the text should be better related in the paragraphs and reordered.

3) In the tables should be written the abbreviated name, as the well-known TNF-a, instead of the full name. Then, it should be specified in abbreviations in the figures text ad whole manuscript. Other examples are Eosinophil cationic protein (ECP), Integrin α4β1 or very late antigen-4 (VLA4), B-cell activating factor (BAFF), among others.

4) I believe that message provided in “Perspectives and Future Directions” and “Conclusions” are mixed and sometimes repeating sentences or information of the main text.

5) The figure 2 should be designed better since it is difficult to see at least in the pdf document. The frame of the cellular compartments and cells blocks name of some metabolites.

I consider that review is clear, deep and with huge information, although to improve the information provided and facilitate to follow the reading of the manuscript, a profound review of the grammar and the way to express the information should be done. The reviewed version that would provide the authors will increase comprehension and reading of the manuscript.

Author Response

The literature analysis was well executed, profound and the “conclusions” (see later comment about that) are consistent with provided data. However, some comments would be done to improve the clearness of the manuscript. Thank you for your comments.

The manuscript is sometimes difficult to be follow in a grammatical point of view. The literature data are exposed very “telegraphically” with one after other reference without, at least for me, a clear line of thinking. Let me refer some examples:

1) Section 1.5 (pg. 3). The use of commas (,) and point and commas (;) to list changes in NMOSD disease is confusing and not “well accepted” as clearest and grammatical use. This way of writing is followed along the manuscript several times. We have reviewed the manuscript and have made the changes as you have suggested.

2) Section 1.6.1 (pg. 4). I believe that for a journal named “Metabolites” is a bit un-sense to give definitions as “metabolism”, “metabolites” or “metabolomics”. May be the overview should be centred (as suggestion) in quality of data provided by metabolomics in comparison with other techniques. Of course, it is a prerogative of the author to decide how to write the manuscript. Thank you for this suggestion - we have deleted/altered this section as you suggested - thank you!

3) In several places of the manuscript are used too “colloquial words or sentences” or “phrase connectors”. For instance, (pg. 8, 2nd paragraph) “With this in mind, six…”, (pg. 8, 3rd paragraph) “…NMOSD. Again, the specificity…” (pg. 9, 3rd paragraph), “…metabolism. Further, given that…”, (pg. 13, 3rd paragraph) “…activity (198-204). As such, due to their…”, (pg. 15, 1st paragraph) “…To begin this exhilarating journey…”, (pg. 16, 1st paragraph) ”…of novel discriminative biomarkers and therapeutic strategies are great”. Thank you for this constructive comment - we have now gone through the entire manuscript and insert many changes throughout to make the text better reflect our intent. We have made a very strong effort to remove the items as you have suggested - we hope that we have been successful. Again, the comment was extremely helpful and we cannot overestimate how much this has improved our manuscript. 

There are also some mistakes that should be corrected:

1) The “supplementary Table 1” or “Table 1” is written in bold letter in some cases and in plain letter in others. This has been corrected.

2) The “supplementary Table 3” is referred in the text BEFORE that “supplementary Table 2”. Then, the message of those tables in the text should be better related in the paragraphs and reordered. This has been corrected.

3) In the tables should be written the abbreviated name, as the well-known TNF-a, instead of the full name. Then, it should be specified in abbreviations in the figures text ad whole manuscript. Other examples are Eosinophil cationic protein (ECP), Integrin α4β1 or very late antigen-4 (VLA4), B-cell activating factor (BAFF), among others. This has been corrected to the extent that we felt comfortable with. Having removed the names from the tables as you had suggested, the tables became to difficult to read - because there was a continual need to go to a list to find out exactly what the abbreviation was. We felt that it was best to have the figures and each table as an independent item - so have not changed these to the extent that you suggested - we do hope that you find this acceptable. 

4) I believe that message provided in “Perspectives and Future Directions” and “Conclusions” are mixed and sometimes repeating sentences or information of the main text. This has been corrected. If fact, we have completely revised these sections.

5) The figure 2 should be designed better since it is difficult to see at least in the pdf document. The frame of the cellular compartments and cells blocks name of some metabolites. This was a technical problem that has been corrected - I hope with the sending of the figures back to the journal in a different formant. Thank you for bringing this to our attention.

I consider that review is clear, deep and with huge information, although to improve the information provided and facilitate to follow the reading of the manuscript, a profound review of the grammar and the way to express the information should be done. The reviewed version that would provide the authors will increase comprehension and reading of the manuscript. Thank you again for your comments - I am pleased with your review.

Reviewer 3 Report

In this Review Manuscript, the Authors aim to provide a comprehensive analysis of metabolomic perturbations and related proteomics outcomes that discriminate NMOSD from other CNS demyelinating inflammatory disorders. In particular, they highlight the potential of short chain fatty acids (SCFA), lipids, lipoproteins, and amino acids as putative biomarkers to discriminate between AQP4-Ab+ NMOSD, MOG-Ab+ NMOSD, and MS, supporting metabolomic profiling as a promising approach for biomarker discovery to improve diagnosis, treatment, and prognosis of NMOSD and related CNS demyelinating autoimmune disorders.

The Manuscript is generally well written well structured and clear.

I have a comment about Chapter 3, pag. 11,  that the Authors named as “Discussion”: I think that it is confusing, considering that at page 14 there is another “Discussion” session. The Authors should clarify this point, I think that paragraphs 3.1 and 3.2 could be included in Session 2 “Metabolomics and NMOSD”.

I have the following suggestions:

  • Introduction:
    • Clinical Characterization: the presence of AQP4 antibodies or MOG antibodies in serum unequivocally allows to differentiate NMOSD from MS or other CNS demyelinating disorders. In addition, other laboratory results such as cerebrospinal fluid (CSF) pleocytosis with eosinophils and/or neutrophils, oligoclonal bands, glial fibrillary acidic protein in the CSF can help neurologists in differentiate NMOSD from other similar disorders. However, many patients, who have overlapping features of NMOSD and other demyelinating disorders, test negative for AQP4 or MOG antibodies and may be difficult to definitively diagnose. This raises important practical issues, since NMO and MS respond differently to immunomodulatory treatment and have different prognoses.

Thus I suggest the Authors to better explain this concept, and to highlight the importance of biomarkers that could discriminate different NMOSD forms in patients negative for AQP4 or MOG antibodies, or biomarkers of response to treatment, or biomarkers of disease prognosis.

1.8 Metabolomics in Multiple Sclerosis: the Authors state that “Metabolomic profiling platforms have successfully validated biomarkers to distinguish between multiple sclerosis and healthy controls, established a metabolomic signature for multiple sclerosis severity, and identified biomarkers to discriminate between RRMS, secondary progressive multiple sclerosis (SPMS), and primary progressive multiple sclerosis (PRMS) (101, 102, 104-123).”

I think that this sentence is too ambitious, the Authors should better explain that the majority of those studies are performed on murine models of MS. Even if a lot of molecules have been shown to be good candidate biomarkers to early discriminate different form of MS (RR, SP or PP), none of them has been “successfully validated” and actually used in clinical practice.                                                    

  • Discussion: page 15. Considering the crucial role of B cells in the pathogenesis of NMOSD, and considering that one of the main current treatment options for NMOSD patients is Rituximab, a monoclonal antibody directed against B cells, should metabolomic profiling be considered an important tool to discriminate between patients that respond and those who don’t respond to an anti-CD20 therapy? Indeed, one of the current challenges in the management of NMOSD patients treated with Rituximab is to early identify non responder patients, avoiding disease relapses.

Author Response

I have a comment about Chapter 3, page. 11, that the Authors named as “Discussion”: I think that it is confusing, considering that at page 14 there is another “Discussion” session. The Authors should clarify this point, I think that paragraphs 3.1 and 3.2 could be included in Session 2 “Metabolomics and NMOSD”. Thank you for your catch - you are correct - we have corrected this oversight.

Clinical Characterization: the presence of AQP4 antibodies or MOG antibodies in serum unequivocally allows to differentiate NMOSD from MS or other CNS demyelinating disorders. In addition, other laboratory results such as cerebrospinal fluid (CSF) pleocytosis with eosinophils and/or neutrophils, oligoclonal bands, glial fibrillary acidic protein in the CSF can help neurologists in differentiate NMOSD from other similar disorders. However, many patients, who have overlapping features of NMOSD and other demyelinating disorders, test negative for AQP4 or MOG antibodies and may be difficult to definitively diagnose. This raises important practical issues, since NMO and MS respond differently to immunomodulatory treatment and have different prognoses.

Thus I suggest the Authors to better explain this concept, and to highlight the importance of biomarkers that could discriminate different NMOSD forms in patients negative for AQP4 or MOG antibodies, or biomarkers of response to treatment, or biomarkers of disease prognosis. Excellent comment - we have made this correction. Thank you.

1.8 Metabolomics in Multiple Sclerosis: the Authors state that “Metabolomic profiling platforms have successfully validated biomarkers to distinguish between multiple sclerosis and healthy controls, established a metabolomic signature for multiple sclerosis severity, and identified biomarkers to discriminate between RRMS, secondary progressive multiple sclerosis (SPMS), and primary progressive multiple sclerosis (PRMS) (101, 102, 104-123).”

I think that this sentence is too ambitious, the Authors should better explain that the majority of those studies are performed on murine models of MS. Even if a lot of molecules have been shown to be good candidate biomarkers to early discriminate different form of MS (RR, SP or PP), none of them has been “successfully validated” and actually used in clinical practice.  We agree with your comment - we have corrected this.                                               

Discussion: page 15. Considering the crucial role of B cells in the pathogenesis of NMOSD, and considering that one of the main current treatment options for NMOSD patients is Rituximab, a monoclonal antibody directed against B cells, should metabolomic profiling be considered an important tool to discriminate between patients that respond and those who don’t respond to an anti-CD20 therapy? Indeed, one of the current challenges in the management of NMOSD patients treated with Rituximab is to early identify non responder patients, avoiding disease relapses. We agree with you - we have made these changes - thank you for your comments.

Round 2

Reviewer 2 Report

I believe authors did a nice effort to improve the “English scientific grammar” to a more direct less colloquial expression of the data.

There are just few comment in relation with the aspect of the manuscript to homogenize the written. For instance, It is written in some cases references in the text as “Author et al” (see lines 369, 553, 555, 638, 651, 657, 772, 774…) and other as  “Author et al year)” (see 356, 369, 362, 367, 374, 377, 560, 561, 566, 769…. An even a case (line 651) Li et. al., Please, homogenize. Also refer with or without bold letter (as recommends the journal) “Supplementary Table… (line 766 vs 840).

Please, double review the whole manuscript to correct those few aspects of the writing.